# Application of Artificial Intelligence in Screening for Adverse Perinatal Outcomes—A Systematic Review

**DOI:** 10.3390/healthcare10112164

**Published:** 2022-10-29

**Authors:** Stepan Feduniw, Dawid Golik, Anna Kajdy, Michał Pruc, Jan Modzelewski, Dorota Sys, Sebastian Kwiatkowski, Elżbieta Makomaska-Szaroszyk, Michał Rabijewski

**Affiliations:** 1Department of Reproductive Health, Centre of Postgraduate Medical Education, Żelazna 90 St., 01-004 Warsaw, Poland; 2Faculty of Medicine, Lazarski University, Świeradowska 43 St., 02-662 Warsaw, Poland; 3Outcomes Research Unit, Polish Society of Disaster Medicine, P.O. Box 78, 05-090 Raszyn, Poland; 4Department Obstetrics and Gynecology, Pomeranian Medical University, Al. Powstańców Wlkp. 72, 70-111 Szczecin, Poland

**Keywords:** artificial intelligence (AI), artificial neuronal network (ANN), pregnancy complications, perinatology

## Abstract

(1) Background: AI-based solutions could become crucial for the prediction of pregnancy disorders and complications. This study investigated the evidence for applying artificial intelligence methods in obstetric pregnancy risk assessment and adverse pregnancy outcome prediction. (2) Methods: Authors screened the following databases: Pubmed/MEDLINE, Web of Science, Cochrane Library, EMBASE, and Google Scholar. This study included all the evaluative studies comparing artificial intelligence methods in predicting adverse pregnancy outcomes. The PROSPERO ID number is CRD42020178944, and the study protocol was published before this publication. (3) Results: AI application was found in nine groups: general pregnancy risk assessment, prenatal diagnosis, pregnancy hypertension disorders, fetal growth, stillbirth, gestational diabetes, preterm deliveries, delivery route, and others. According to this systematic review, the best artificial intelligence application for assessing medical conditions is ANN methods. The average accuracy of ANN methods was established to be around 80–90%. (4) Conclusions: The application of AI methods as a digital software can help medical practitioners in their everyday practice during pregnancy risk assessment. Based on published studies, models that used ANN methods could be applied in APO prediction. Nevertheless, further studies could identify new methods with an even better prediction potential.

## 1. Introduction

The first mentions of artificial intelligence (AI) appeared in ancient Greek mythology. In the 15th century, Leonardo Da Vinci crafted an impressive mechanical knight that could move its arms, sit, and twist its head. However, the actual field of artificial intelligence research was founded in 1956 [1].

Artificial intelligence is a science and engineering field that deals with computer-driven mechanisms or machines that use computer power, memory, and large amounts of data. It has been handling learning tasks by creating models of intelligent behavior with the lowest possible human interruption. The appliance of artificial intelligence in medicine can be divided into two main branches: virtual and physical. The virtual branch is represented by machine learning (ML) which is characterized by computer algorithms that can enhance learning through experience [1]. In contrast, the physical branch consists of mechanical devices such as medical equipment or highly advanced robots [2].

Many complications of pregnancy cannot be directly treated once they occur. Therefore, perinatal medicine predicts high-risk groups and applies interventions to minimize adverse perinatal outcomes [3]. Examples of such interventions are aspirin for preeclampsia, progesterone for preterm birth, diet and insulin for diabetes, and growth assessment for stillbirth [4,5,6,7,8,9]. Incorporating AI in risk assessment and adverse pregnancy outcome (APO) prediction is an opportunity for present-day perinatal medicine. The development of prediction models and effective screening methods for APO could improve diagnostic and therapeutic decisions and significantly impact women’s and children’s well-being. The routine implementation of AI could even further improve medical care [10,11,12,13].

Factors such as BMI, maternal hypertension, CTG, and gestational diabetes mellitus can help predict fetal disorders, the route of delivery, preterm birth, and other risk factors for pregnant women and fetuses. In addition, maternal biochemical markers, such as β-human chorionic gonadotrophin (β-hCG), pregnancy-associated plasma protein A (PAPP-A), and placental growth factor (PLGF), are used to predict aneuploidies or other fetal and placental dysfunctions [14,15]. Artificial neural networks (ANNs) are the most popular form of AI applied in medicine. ANNs are computer analytic models inspired by the animal nervous system. An ANN is composed of interconnected computer processors called ‘neurons’ which can simultaneously perform data calculations to assess the occurrence of an outcome based on given factors. ANNs can learn and analyze imprecise pieces of information, nonlinear data, and previous examples. These abilities allow for the analysis of large amounts of medical data and help in diagnostic and treatment decisions. They progressively move towards plastic neural systems capable of learning through evolution by interacting with the environment [16]. The other most popular AI techniques are ML, the decision tree (DT), the support vector machine (SVM), and the random forest (RF) [17,18,19,20,21,22]. Every AI method should be compared to the best prediction statistic method which is named logistic regression (LR), or, to be precise, a distinguished form of multiple linear regression—multivariate logistic regression (MLR). MLR has the same aim as all model-building techniques: to describe the relationship between predictors and outcomes [23,24,25].

There are no published studies comparing the usage of artificial intelligence methods to predict perinatology complication occurrences. This study investigated the evidence for applying artificial intelligence methods in obstetric pregnancy risk assessment and adverse pregnancy outcome prediction.

## 2. Materials and Methods

### 2.1. Study Design

The current systematic review was performed according to the systematic reviews and meta-analyses (PRISMA) guidelines and was presented in Appendix A [26]. The systematic review protocol was registered in PROSPERO with ID-number CRD42020178944 and was published beforehand [27].

### 2.2. Search Strategy

Authors screened the following databases using the search strategy presented in Box 1: Pubmed/MEDLINE, Web of Science, Cochrane Library, and EMBASE. In addition, the electronic database search was supplemented by searching Google Scholar.

Box 1Search strategy.(pregnant OR pregnancy OR prepartum OR prenatal OR gestation OR prelabor OR maternal) AND (artificial neural networks OR artificial intelligence OR machine learning) AND (pregnancy risk)

All searches were conducted on 27 May 2022, with languages restricted to English, German, or Polish and with no publication-time limits. Additionally, references of all included studies were hand-searched for additional relevant articles.

### 2.3. Inclusion Criteria

All types of evaluative study designs were included and assessed. Two reviewers (SF and MP) independently screened the studies by the title, abstract, and full text. Studies that met the selection criteria were included. The reference lists of the included studies were additionally screened. Every included study was assessed (0 = not relevant, 1 = possibly relevant, and 2 = very relevant). Only publications that scored at least 1 point were included in the study. Any disagreement was discussed and resolved by the third researcher (AK).

### 2.4. Data Extraction

PI(E)CO question was “Artificial intelligence methods in the screening for pregnancy risk and adverse pregnancy outcomes in pregnant women.” Population (P) was pregnant women with high-risk pregnancies (with named complications of pregnancies). Intervention (E) applied artificial intelligence methods to evaluate pregnancy risk and to screen for APO. For comparison, (C) pregnant women with low-risk pregnancies were included. The outcome (O) was the prediction value of the studied artificial intelligence method. Studies (S) included in the analyses were retrospective or prospective trials with the unaffected population as a control.

The PRISMA diagram was made according to Reporting Items for Systematic Reviews and Meta-Analyses. The PRISMA flow chart is shown in Figure 1 [28].

### 2.5. Quality Assessment and Risk of Bias

The risk of bias was assessed independently by two authors (SF and MP) using the Newcastle–Ottawa scale [29]. The third reviewer (AK) resolved apparent discrepancies in the selection process. In general, the studies included were of moderate to high quality. The selection process is shown in Table 1.

### 2.6. Synthesis of Results

Due to the heterogeneity of the included studies, there was no possibility of performing a quantitative synthesis. Nevertheless, all prediction values of the included studies’ AI methods were divided into groups according to pregnancy outcomes and were assessed. Results were summarized in Table 2, Table 3, Table 4, Table 5, Table 6, Table 7, Table 8, Table 9 and Table 10.

**Table 1 healthcare-10-02164-t001:** The Newcastle–Ottawa scale for quality assessment of included studies.

		Authors																																																
	**Item**	Gorthi et al. (2009)	Umoh and Nyoho (2015)	Fernandes et al. (2017)	Chaminda and Sharmilan (2016)	Moreira et al. (2018)	Neocleous et al. (2017)	Neocleous et al. (2016)	Neocleous et al. (2018)	Akbulut et al. (2018)	Robinson et al. (2010)	Moreira et al. (2018)	Nair et al. (2018)	Jhee et al. (2019)	Mello et al. (2001)	Li et al. (2016)	Kuhle et al. (2018)	Moreira et al. (2019)	Naimi et al. (2018)	Kayode et al. (2016)	Harihara et al. (2019)	Koivu et al. (2020)	Malacov et al. (2020)	Shanker et al. (1996)	Polak and Mendyk (2004)	Artzi et al. (2020)	Moreira et al. (2018)	Nanda et al. (2011)	Kang et al. (2019)	Pourahmad et al. (2017)	Weber et al. (2018)	Idowu et al. (2015)	Woolery and Jerzy (1994)	Fergus et al. (2013)	Courtney et al. (2008)	Nodelman et al. (2020)	Moreira et al. (2018)	Bahado-Singh et al. (2019)	Lee and Ahn (2019)	Goodwin et al. (2001)	Prema and Pushpalatha (2019)	Elaveyini et al. (2011)	Catley et al. (2006)	Beksac et al. (1996)	Caruana et al. (2003)	Paydar et al. (2017)	Li et al. (2017)	Grossi et al. (2016)	Valensise et al. (2006)	Gao et al. (2019)
A	**Selection**																																																	
	Exposed were truly representative of average	1	0	1	1	1	1	1	1	0	1	1	0	1	1	1	1	1	1	1	1	1	1	1	1	1	1	1	1	1	1	1	1	1	1	1	1	0	1	1	1	0	1	1	1	1	1	0	1	1
	Selection of nonexposed from the same community	1	0	1	1	1	1	1	1	0	1	1	0	1	1	1	1	1	1	1	1	1	1	1	1	1	1	1	1	1	1	1	1	1	1	1	1	0	1	1	1	1	1	1	1	1	1	0	1	1
	Exposure of ascertained by secure record or interview	0	0	1	1	1	1	1	1	1	1	1	1	1	1	1	1	1	1	1	1	1	1	1	1	1	1	1	1	1	1	1	1	1	1	1	1	1	1	1	1	1	1	0	1	1	1	1	1	1
	Demonstration of outcome of interest not present at the start of the study	1	1	1	1	1	1	1	1	1	1	1	1	1	1	1	1	1	1	1	1	1	1	1	1	1	1	1	1	1	1	1	1	1	1	1	1	1	1	1	1	1	1	1	1	1	1	1	1	1
B	**Comparability**																																																	
	Study controls for other variables	1	1	1	1	1	1	1	1	1	1	1	1	1	1	1	1	1	1	1	1	1	1	1	1	1	1	1	1	1	1	1	1	1	1	1	1	1	1	1	1	1	1	1	1	1	1	1	1	1
C	**Outcome**																																																	
	Follow up long enough for outcome to occur	1	0	0	0	1	0	0	0	0	0	0	0	0	0	0	0	0	0	0	0	0	0	0	0	0	0	0	0	0	0	0	0	0	0	0	0	0	0	0	0	0	1	0	0	1	1	0	0	0
	Complete follow up of all subjects accounted for	1	0	0	0	1	0	0	0	0	0	0	0	0	0	0	0	0	0	0	0	0	0	0	0	0	0	0	0	0	0	0	0	0	0	0	0	0	0	0	0	0	1	0	0	1	1	0	0	0
	Subject to follow up, unlikely to introduce biases	1	0	0	0	1	0	0	0	0	0	0	0	0	0	0	0	0	0	0	0	0	0	0	0	0	0	0	0	0	0	0	0	0	0	0	0	0	0	0	0	0	1	0	0	1	1	0	0	0
	Assessment of outcomes	0	0	0	0	1	0	0	0	1	1	1	0	1	0	1	1	1	1	0	0	1	1	1	0	0	1	0	1	0	0	1	0	1	1	0	0	0	1	0	0	0	0	0	0	0	0	0	0	0
	**Score**	7	2	5	5	9	5	5	5	4	6	6	3	6	5	6	6	6	6	5	5	6	6	6	5	5	6	5	6	5	5	6	5	6	6	5	5	3	6	5	5	4	8	4	5	8	8	3	5	5

**Table 2 healthcare-10-02164-t002:** Characteristics of included studies on general pregnancy risk assessment.

Study	Country	Sample Size	Population	Model	Knowledge Source for AI	Type of Risk	Prediction Value of AI/AUC	Prediction Value of LR
Gorthi et al. (2009) [30]	India	240	A prospectively collected sample of pregnant women was used to assess the practical model. There were 200 training cases and 40 test cases.	Knowledge-based system	Literature	Risk classification	Training 93.4 %, test 82.5%	NA
Umoh and Nyoho (2015) [31]	Nigeria	30	Pregnant women (aged 25–40) were selected to test the theoretical model.	Intelligent fuzzy framework	Literature	High-risk pregnancy	Not assessed	NA
Fernandes et al. (2017) [32]	Brazil	1380	Retrospective validation of the documentation of pregnant women from the High-Risk Prenatal sector at MEJC was used to test the theoretical model.	Knowledge-based system	Predefined risk factors	Risk reclassification	Not assessed	NA
Chaminda and Sharmilan (2016) [33]	Sri Lanka	117	Pregnant women of different ages and lifestyles were used. (Unclear if retrospective or prospective.) There were 93 training cases and 24 testing cases.	Hybrid system: neuronal network and naïve Bayes algorithm	Predefined risk factors	Pregnancy risk assessment	ANN 80%, naïve Bayes 70%, novel hybrid approach 86%	NA
Moreira et al. (2018) [34]	Brazil, Portugal, Saudi Arabia, India, Russia	100	Parturient women diagnosed with a hypertensive disorder during pregnancy were used. All prospectively collected cases were used to test the model.	Artificial neural networks (ANN)	Patient’s history	Hypertensive disorder during pregnancy	Hybrid algorithm 93%	NA

## 3. Results

AI has been applied to many different aspects of perinatal medicine. The included studies described AI applications in predicting general pregnancy risk characteristics, prenatal diagnoses, pregnancy hypertension disorder, fetal growth, stillbirth, gestational diabetes, preterm deliveries, the delivery route, and other conditions [30,31,32,33,34,35,36,37,38,39,40,41,42,43,44,45,46,47,48,49,50,51,52,53,54,55,56,57,58,59,60,61,62,63,64,65,66,67,68,69,69,70,71,72,73,74,75,76,77,78,79]. The studies were divided into nine groups using the predictive value of the AI methods. Table 3, Table 4, Table 5, Table 6, Table 7, Table 8, Table 9 and Table 10 present the main characteristics of the included studies. For each group, the most robust method was identified.

The first group included five prediction studies assessing the general pregnancy risk (Table 2) [30,31,32,33,34].

There were various predictors used in the model construction. The most relevant were the clinical parameters and existing health conditions of the mother (maternal and gestational age; gravidity and parity; BMI, weight gain; and life parameters such as blood pressure, lifestyle, nutrition, etc.), pregnancy-related complications (threatened preterm delivery, bleeding in pregnancy, prepregnancy and gestational diabetes, hypertensive disorder spectrum, HELLP syndrome, cholestasis and other liver disorders in pregnancy, thrombophilia, autoimmune diseases, an age higher than 35 years, multiple pregnancies, and an interval of 10 years or more between pregnancies), laboratory parameters (albuminuria, hyperglycemia, leucocythemia, etc.), fetal monitoring parameters (basal fetal heart rate, variability, and the occurrence of accelerations or decelerations), and Ultrasound and Doppler parameters (fetal movement; the growth of the fetus; uterine, cerebral, and umbilical fetal Doppler signals; and the amniotic fluid index). However, many other variables might be used in order to measure the overall risk of a pregnancy. This would increase the model’s representativeness and accuracy, but it could be difficult to determine how effective the model would be.

Classification and regression trees (CART) [30], fuzzy logic [31], the teletech architecture ILITIA [32], ANNs, and naïve Bayes were used [34]. CART had the best predictive value (accuracy of 93.4% in the training group and 82.5% in the tested group), and naïve Bayes had the worst accuracy with a score of 70% [34]. Nevertheless, the observed differences between all the methods were minimal.

The second group included four articles about the prenatal diagnosis of chromosomal abnormalities using ANNs and binary classification models such as the averaged perceptron, the boosted DT, the Bayes point machine, the decision forest, the decision jungle, the locally deep SVM, LR, ANNs, and the SVM (Table 3) [35,36,37,38].

**Table 3 healthcare-10-02164-t003:** Characteristics of included studies on prenatal diagnosis of chromosomal abnormalities.

Study	Country	Sample Size	Population	Model	Knowledge Source for AI	Type of Risk	Prediction Value of AI/AUC	Prediction Value of LR
Neocleous et al. (2016) [36]	United Kingdom, Netherlands, Cyprus	51,208	Pregnant women with euploid and aneuploid fetuses were used. There was a 3-fold cross validation of the system. This was done by randomly dividing cases into training and evaluation groups.	ANN	Patient’s history	Aneuploidies	21st trisomy 100%, euploidy 96.1%	NA
Neocleous et al. (2017) [35]	United Kingdom, Netherlands, Cyprus	123,329	There were 122362 euploid cases and 967 aneuploid cases. There were retrospective cases of pregnant women. They were split into 70% training cases and 30% validation cases.	ANN	Patient’s history	Aneuploidies	21st trisomy 100%, other aneuploidies > 80%	NA
Neocleous et al. (2018) [37]	United Kingdom, Netherlands, Cyprus	72,654	There was a prospective sample of pregnant women at 11–13 weeks gestation. An amount of 70% of training sets and 30% of test sets were randomly chosen.	ANN	Patient’s history	Aneuploidies	21st trisomy 94.2%, other aneuploidies 79.5%	NA
Akbulut et al. (2018) [38]	Turkey, United States	97	There was a prospective analysis of pregnant women (96 singletons and 1 twin). There were 97 training cases and 16 testing cases.	Decision Forest (DF)	Maternal questionnaire, specialist, and patient’s history.	Congenital anomalies	DF during training 89.5%, DF during testing 87.5%	NA

A well-known chromosomal abnormality prediction was built in the AI model using ultrasound indicators (nuchal translucency, crown-rump length, and the presence of the nasal bone) and pregnancy-associated plasma protein A (PAPP-A) with free β-hCG. All were obtained between the 9th + 3 and the 11th week + 6 days of gestation.

The best method was the Decision Forest model, which achieved the highest accuracy of 89.5%, with an almost 100% detection rate of 21st trisomy [38].

The third group included five articles about pregnancy hypertension characteristics (Table 4) [39,40,41,42,43]. The methods used in this group were ANNs [41], neuro-fuzzy machine learning techniques [40], MLR [39,43], and other ML techniques such as LR, the DT model, the naïve Bayes classification, the SVM, the RF algorithm, and the stochastic gradient boosting method [42].

**Table 4 healthcare-10-02164-t004:** Characteristics of included studies on hypertension.

Study	Country	Sample Size	Population	Model	Knowledge Source for AI	Type of Risk	Prediction Value of AI/AUC	Prediction Value of LR
Robinson et al. (2010) [39]	United States	608	There was a retrospective analysis of patients with preeclampsia (after induction of labor, 1997–2007, 195 cesarean sections, 413 vaginal deliveries). There was training of 304 patients (50%) and testing of 152 patients (25%).	ANN	Patient’s history	Preeclampsia	AUC (area under the ROC curve) 0.75	AUC 0.74
Moreira et al. (2018) [40]	Brazil, Portugal, Russia, Saudi Arabia	205	There were 205 women with a hypertensive disorder during pregnancy and 7 women with HELLP syndrome. All records were used to test the model.	Neuro-fuzzy model	Patient’s history, experts	HELLP syndrome	AUC 0.685	NA
Nair et al. (2018) [41]	United States	38	There were 38 pregnant women (19 with PE and 19 normotensives) It was split into 85 training cases and 15% testing cases.	ANN	Patient’s history	Preeclampsia	AUC 0.908	NA
Jhee et al. (2019) [42]	Korea	11,006	There was a prospective analysis of pregnant women. It was split into 70% (n = 10 058) training cases and 30% (n = 474) testing cases.	ML—decision tree (DT), naïve Bayes classification (NBC), support vector machine (SVM), RF, stochastic gradient boosting (SGB)	Patient’s history	Preeclampsia	DT 84.7%, NBC 89.9%, SVM 89.2%, RF 92.3%, SGB 97.3%	86.2%
Mello et al. (2001) [43]	Italy	303	There was a prospective analysis of preconception enrollment and, consequently, pregnant women (spontaneous conception, single pregnancies, 76–25.1% with pregnancy hypertension in III trimester)—patients were postpartum controlled. There were 187 training cases and 116 testing cases.	ANN and multivariate logistic regression (MLR)	Patient’s history	Preeclampsia and FGR	AUC 0.952, positive predictive value 86.2%, negative predictive value 95.5%	AUC 0.962, positive predictive value 92%, negative predictive value 93.4%

For pregnancy hypertension prediction, the following risk factors were assessed: maternal age at the time of delivery, gestational age at delivery; maternal race; parity; neonatal birth weight; prepregnancy BMI; cervical ripening during induction; fetal growth restriction; blood pressure; maternal medical history of hypertension, diabetes, and previous preeclampsia; obstetrical and social histories; medications prescribed during pregnancy; and laboratory data (blood urea nitrogen, serum creatinine, spot urine protein-to-creatinine ratio, urine albumin-to-creatinine ratio, hemoglobin, fasting blood glucose, serum albumin, uric acid, total bilirubin, aspartate transaminase, alanine transaminase, total cholesterol, triglycerides, high-density lipoprotein cholesterol, and low-density lipoprotein cholesterol).

ML techniques showed remarkable results. The stochastic gradient boosting model had the best prediction performance, with an accuracy of 97.3% and a false-positive rate of 0.9%. ANNs and MLR showed an area under the ROC curve of 0.952 as well as 86.2% sensitivity and 95.4% specificity for ANNs.

The fourth group included four articles about the prediction of fetal growth restriction (FGR) (Table 5) [44,45,46,47].

**Table 5 healthcare-10-02164-t005:** Characteristics of included studies on fetal growth.

Study	Country	Sample Size	Population	Model	Knowledge Source for AI	Type of Risk	Prediction Value of AI/AUC	Prediction Value of LR
Li et al. (2016) [44]	China	215,568	There was a prospective analysis of pregnant women (13,258 cases of SGA and 202310 cases of non-SGA). It was split into 90% training cases and 10% testing cases. It was unclear if they were prospective or retrospective.	ML—support vector machine (SVM), random forest (RF), logistic regression (LR), and sparse LR	Patient’s history	Fetal growth abnormalities	SVM 92.4%, C4.5 43.7%, RF 61.2%, LR Sparse 94.5%, AUC 0.6	93%, AUC 0.6
Kuhle et al. (2018) [45]	Canada	30,705	There was a retrospective analysis of pregnant women after 26 gestation weeks (SGA 7.9%, LGA 13.5%). It was split into 80% training cases and 20% testing cases.	Neural network models: NNET package	Patient’s history	Fetal growth abnormalities	AUC 0.60–0.75	84.7%, 0.66
Moreira et al. (2019) [46]	Brazil	NA	There was a prospective analysis of pregnant women (Fetal birth-weight estimation in high-risk pregnancies). It was not possible to assess the size of the training and test groups.	Machine learning (ML)—bagged tree	NA	Fetal Growth	84.9%, AUC 0.636	NA
Naimi et al. (2018) [47]	United States	18,757	There was a retrospective analysis of pregnant women (240 high-risk pregnancies). All cases were used to test the model.	ML	Patient’s history	Fetal Growth	Not assessed	NA

The three most important factors for SGA prediction were gestational weight gain, maternal smoking, and prior low birth weight (LBW) infants. Prepregnancy BMI, gestational weight growth, and a history of deliveries of infants weighing more than 4080 g were the primary predictors of LGA.

Methods such as the SVM, the RF, LR, sparse LR models, linear and quantile regression, Bayesian additive regression trees, generalized boosted models, and the ML technique called the bagged tree were used [44,45,46,47]. The SGA bagged tree case had a prediction value of 84.9% and an area under the receiver operating characteristic curve of 0.636 [46]. The highest accuracy was achieved by the SVM, with a prediction score of 90.7% and an AUC of 0.588 [44].

The next group of examined articles included four studies about stillbirth prediction (Table 6) [48,49,50,51]. MLR [48], artificial intelligence analysis of time-lapse (TLM) embryo images [49], LR, ANNs, the gradient boosting DT [50], regularized logistic regression, the DT based on classification and regression trees, the RF, extreme gradient boosting (XGBoost), and the multilayer perceptron neural network were used in those studies [51].

**Table 6 healthcare-10-02164-t006:** Characteristics of included studies on stillbirth.

Study	Country	Sample Size	Population	Model	Knowledge Source for AI	Type of Risk	Prediction Value of AI/AUC	Prediction Value of LR
Kayode et al. (2016) [48]	Nigeria, Netherlands, Ghana, South Africa	6956	There was a retrospective analysis of pregnant women (6573 well-ended pregnancies and 443 stillbirths). All cases were used as testing cases.	Multivariable logistic regression	Patient’s history	Stillbirth	C-statistic basic model 80% Extended model 82%	NA
Harihara et al. (2019) [49]	United Kingdom, Spain, Italy, Brazil, United States	3412	Images of the embryos (1756 newborns, 1656 miscarriages) were used. A total of 63% (n = 2140) of retrospective lapse images of blastocysts with known live-birth outcomes following a single embryo transfer were used to train the model. An amount of 15.5% (n = 536) of the images were used in validation. A total of 21.5% (n = 736) of prospective cases were used to test the model.	Convolutional neural network (CNN)	Patient’s history	Miscarriage	77%	NA
Koivu et al. (2020) [50]	Finland	16,340 661	Prospectively collected normal pregnancies (965,504 preterm births, 8061 early stillbirths, and 8420 late stillbirths) were used. There were 9,004,902 training cases and 1,292,847 testing cases.	LR, ANNs, deep NN (neuronal network), SELU N (scaled exponential linear units network), LGBM (The lightgun gradient boosting decision tree)	Patient’s history	Early and late stillbirth, preterm delivery	Early stillbirth AUC Deep NN 0.73, SELU N 0.75, LGBM 0.75; Late stillbirth AUC Deep NN 0.57, SELU N 0.59, LGBM 0.6; Preterm delivery AUC Deep NN 0.66, SELU N 0.67, LGBM 0.67	Early stillbirth AUC 0.73, Late stillbirth AUC 0.58, Preterm delivery AUC 0.64
Malacova et al. (2020) [51]	Australia, Norway, United States	467,365	There was a retrospective analysis of pregnant women (7788 stillbirths). All cases were used for testing the existing models.	LR, decision trees (DT) and regression trees, random forest (RF), extreme gradient boosting (XGBoost), and a multilayer perceptron neural network	Patient’s history	Stillbirth	AUCs 0.59–0.84, DT 0.59–0.82, RF 0.594–0.84, XGBoost 0.596–0.84, multilayer perceptron 0.595–0.84	AUCs 0.602–0.83

Social predictors (maternal age, parity, education, occupation, ethnicity, place of residence, previous fetal loss, bleeding during pregnancy, maternal BMI, number of prior caesarean sections, multiple pregnancies, child’s gender, and fetal growth rate) and comorbid conditions (hypertension disorder spectrum, diabetes, sickle cell disease, renal disorders, thyroid dysfunction, and venereal diseases) were used to estimate the risk of miscarriages. MLR showed excellent results with a C-statistic basic model of 80% of stillbirth predictions.

The sixth group included seven articles about gestational diabetes prediction (Table 7) [52,53,54,55,56,57,58]. The methods used were ANNs, LR, ML methods, the radial basis function network (RBFNetwork), the MLP, the DT, and the SVM [52,53,54,55,56,57,58].

Other factors such as the mother’s age, BMI, triceps skin-fold thickness, plasma glucose concentration at 2 h in an oral glucose tolerance test, 2 h serum insulin level, and diabetes degree function were particularly effective in predicting gestational diabetes. The ML method resulted in a high accuracy even at pregnancy initiation with 0.85 ROC, substantially outperforming a baseline risk score of 0.68 ROC [52]. The usage of the RBFNetwork also showed great results with a precision of 78.5%, an F-Measure of 78.6%, an ROC area of 0.839, and a Kappa statistic of 0.509 [53]. Therefore, these two models seem to be the best fit for the prediction of GDM [52,53].

**Table 7 healthcare-10-02164-t007:** Characteristics of included studies on gestational diabetes.

Study	Country	Sample Size	Population	Model	Knowledge Source for AI	Type of Risk	Prediction Value of AI/AUC	Prediction Value of LR
Shanker et al. (1996) [55]	United States	768	Pima Indian pregnant women (268 with diabetes) were used. There were 576 training cases (378 patients without diabetes and 198 patients with diabetes) and 192 test cases.	ANN	Patient’s history	GDM	77.6%	79.2%
Polak and Mendyk (2004) [56]	Poland	2551	There was a retrospective analysis of pregnant women (2460 without GDM, 91 with GDM) The randomly chosen 90% were used for training, and the remaining 10% were used to test the model.	ANN	Patient’s history	GDM	70%	56.3%
Artzi et al. (2020) [52].	Israel	588,622	A retrospectively collected cohort of pregnant women (I group—46,002 women from Jerusalem; II group—8540 women from the aforementioned area) was used. There were 2355 training cases (295 at the start of the pregnancy, 2060 generated by different processes). The hold-out/external validation n = 82678.	ML	Patient’s history	GDM	AUC 0.85, AUC 0.80 (simpler model)	NA
Moreira et al. (2018) [53]	USA (Gila and Salt rivers)	394	A prospectively collected cohort of pregnant women (the Pima Indians) was used. All cases were used to test the model.	ANN—multilayer perceptron (MLP)	Literature, patient’s history	GDM	Precision 0.785, F-measure 0.786, AUC 0.839	NA
Nanda et al. (2011) [58]	United States	11,464	There was a retrospective analysis of pregnant women (297 (2.6%) with GDM and 11,167 without GDM). All cases were used to test the model.	Knowledge-based system	Patient’s history	GDM	61.6%	NA
Kang et al. (2019) [57]	China, United States	1891	There was a retrospective analysis of pregnant women with GDM (14.2%, n = 268) that had macrosomia. There were 1702 training cases and 189 test cases.	Decision tree (DT), support vector machine (SVM), and ANN	Patient’s history	Macrosomia in patients with GDM	DT training 87.14, DT test 86.25, ANN training 86.54, ANN test 85.52, SMV training 86.23, SMV test 86.09	Training 86.44, test 86.20

The seventh group included fourteen studies about preterm delivery prediction (Table 8) [59,60,61,62,63,64,65,66,67,68,69,70,71,72]. ANNs [59], cross-validated regressions [60], the RF classifier, the rule-based classifier, penalized logistic regression [61], the synthetic minority oversampling technique [63], LR, classification and regression trees (CART), the SVM, the Bayesian classifier [64], the back-propagation neural network [65], the system for mobile health SVM [72], deep learning [66], data mining, and the feed-forward back-propagation network were used in all of these studies [67,68,69,70,71].

**Table 8 healthcare-10-02164-t008:** Characteristics of included studies on preterm delivery.

Study	Country	Sample Size	Population	Model	Knowledge Source for AI	Type of Risk	Prediction Value of AI/AUC	Prediction Value of LR
Pourahmad et al. (2017) [59]	Iran	1102	There was a retrospective analysis of pregnant women (1047 (95%) singleton pregnancies, 52 (4.7%) employed women, 24.3% of fetuses with PTB). All cases were used to test the model.	ANN	Patient’s history	Preterm delivery (PTB), low birth weight (LBW)	For PTB 81.4% (AUC 0.78), for LBW 87.8% (AUC 0.79)	NA
Weber et al. (2018) [60]	United States	336,214	There was a retrospective analysis of pregnant women (singleton pregnancies, nulliparous, NH black (54,084), and NH white (282,130)). The original sample was partitioned into a training set to fit the model and a testing set to evaluate the goodness of the fit.	ML	Patient’s history	Preterm delivery	AUC 0.67	NA
Idowu et al. (2015) [61]	United Kingdom	300	There was an analysis of pregnant women (262 delivered at term and 38 prematurely). (NA if retrospective or prospective.) All cases were used as a testing group.	ML—RF	Patient’s history	Preterm delivery	AUC 0.94	NA
Woolery and Jerzy (1994) [62]	United States	9419	A population of pregnant women was used for testing the model. (NA if retrospective or prospective.)	ML	Patient’s history (database)	Preterm delivery	53–88%	NA
Fergus et al. (2013) [63]	United Kingdom	300	There was a retrospective analysis of pregnancies (38 ended preterm, and 262 were term deliveries). A total of 80% of the whole dataset was designated for training, and the remaining 20% was for testing.	ML	Patient’s history	Preterm delivery	AUC 0.95	NA
Courtney et al. (2008) [64]	United States	73,040	There was a retrospective analysis of pregnant women. All cases were used to test the method.	Matlab^®^ Neuronal Networks package and the support vector machine (SVM) classifier	Patient’s history, medical records	Preterm delivery, low birth weight	AUC neural networks 0.57, AUC SVM 0.57, AUC Bayesian classifiers 0.59, AUC CART 0.56	AUC 0.605
Nodelman et al. (2020) [65]	United States	3001	There was a retrospective analysis of pregnant women (10.3% preterm deliveries). There was a total of 2038 training cases and 963 testing cases.	ANN	Patient’s history	Preterm delivery	87.3% (95%CI: 85.1–89.4%)	NA
Moreira et al. (2018) [72]	Brazil, Portugal, Spain	205	Pregnant women with a hypertensive disorder during pregnancy were collected retrospectively (12% PTB). Patients were divided into ten subsets of equal sizes. Then, each subset was used once for testing, and the remaining were used for training. (NA if retrospective or prospective.)	Data mining, ML—support vector machine (SVM).	Patient’s history	Preterm delivery in patients with hypertensive disorder	82.1% (AUC 0.785)	NA
Bahado-Singh et al. (2019) [66]	United States	32	We retrospectively collected a cohort of pregnant women (42.3% (n = 11) patients delivered ≥ 34 weeks, 57.7% (n = 15) delivered < 34 weeks). We randomly split the combined omics sample data into 80% training set and a 20% test set.	ML	Patient’s history	Preterm delivery	AUC 0.875	NA
Lee and Ahn (2019) [67]	Korea	596	A cohort of pregnant women was collected retrospectively. There were 298 training cases and 298 validation cases.	ANN	Patient’s history	Preterm delivery	MLR 0.918, Decision Tree 0.8328, naïve Bayes 0.1115 RF 0.8918 SVM 0.9148	NA
Goodwin et al. (2001) [68]	United States	19,970	Pregnant women (105 (1%) American Indian or Alaskan native; 116 (1%) Asian or Pacific Islander; 10,901 (55%) Black not of Hispanic Origin; 519 (3%) Hispanic; 7837 (39%) White not of Hispanic origin; 492 (2%) Unknown) were used (probably retrospectively collected). All cases were used to test the model.	Data Mining	Patient’s history	Preterm delivery	ROC neural net 0.64, custom classifier software 0.72	0.66
Prema and Pushpalatha (2019) [69]	India	124	Preterm birth in pregnant women with diabetes mellitus or gestational diabetes mellitus was used. All cases were used to test the model.	ML, support vector machine (SVM)	Patient’s history	Preterm delivery, DM, GDM	86%	NA
Elaveyini et al. (2011) [70]	India	50	A prospective cohort of pregnant women was used. There were 40 training cases and 10 testing cases.	Neural networks	Patient’s history	Preterm delivery	70%	NA
Catley et al. (2006) [71]	Canada	19710	A prospective cohort of pregnant women, collected before 23 weeks of gestation, was used. They were split into 2/3 training cases and 1/3 test cases.	ANNs	Patient’s history	Preterm delivery	Eight-node high-risk PTB model 0.73 four-node high-risk PTB model with free-flow oxygen cases removed 0.72	NA

Predictors such as first trimester bleeding, preterm rupture of the membranes, polyhydramnios, oligohydramnios, the occurrence of infections, close spacing between pregnancies, histories of preterm delivery, and miscarriages played a crucial role in preterm delivery despite the frequently used clinical evaluation of patients and their comorbidities.

Out of all AI methods used, the RF performed well, with a sensitivity of 97%, a specificity of 85%, an area under the ROC curve of 94%, and a mean square error rate of 14% [61]. Prediction values for most of the studies were similar.

The eighth group includes two articles about predicting the delivery route (Table 9) [73,74]. The DT [73] and the backpropagation learning algorithm (ANN method) [74] were used.

**Table 9 healthcare-10-02164-t009:** Characteristics of included studies on delivery routes.

Study	Country	Sample Size	Population	Model	Knowledge Source for AI	Type of Risk	Prediction Value of AI/AUC	Prediction Value of LR
Beksac et al. (1996) [74]	Turkey	7398	There were retrospective analyses of pregnancies. There were 4451 (40.2%) training cases and 2947 (39.8%) testing cases.	ANN	Literature	Caesarian section rate	Positive predictive value 81.8%, negative predictive value 93.1%	NA
Caruana et al. (2003) [73]	United States	22,175	A sample of pregnant women was used to test the model (probably retrospective).	MML decision trees	Patient’s history	Caesarian Section	Test 87% (AUC 0.9233)	NA

The predictors included maternal age, gravida, parity, gestational age at delivery, need for and kind of labor induction, baby’s presentation at birth, and maternal comorbidities. Again, ANNs performed better than ML, with a 97.5% specificity and a sensitivity of 60.9% [74].

The last group consists of five studies predicting other aspects of perinatal care (Table 10) [75,76,77,78,79].

**Table 10 healthcare-10-02164-t010:** Characteristics of other included studies.

Study	Country	Sample Size	Population	Model	Knowledge Source for AI	Type of Risk	Prediction Value of AI/AUC	Prediction Value of LR
Paydar et al. (2017) [75]	Iran	149	There was a retrospective analysis across pregnant women with systemic lupus erythematosus. For MLP (neuronal network-multi-layer perceptron), 70% were training cases, 15% were validation cases, and 15% were testing data. For RBF (radial basis function), 70% were training cases and 30% were testing data.	CDSS (MLP, RBF)	Literature, specialist, patient’s history	High-risk pregnancy- (with SLE)	RBF 75.16%, MLP 90.6%	NA
Li et al. (2017) [76]	China	358	There was a case-control study of pregnant women (119 fetuses with congenital heart disease and 239 controls). It was split into 85%, or n = 300 (101 cases and 199 controls), training cases and 15%, or n = 58 (18 cases and 40 controls), test cases.	ANN − ANN + BPNN (backpropagation neural network)	Patient’s history, specialist	Congenital heart disease	Training 91%, testing 86%	NA
Grossi et al. (2016) [77]	Italy	137	There was a retrospective analysis of pregnant women (45 mothers of autistic children and 68 mothers of typically developing children). A total of 24 siblings of 19 autistic children were an internal control group. All cases were used to test the model.	Specialized ANNs (ANNs)	Interview of the mothers, literature, patient’s history	Autism	80.2%	46%
Valensise et al. (2006) [78]	Italy	302	There was a retrospective analysis of healthy post-term pregnancies (42 fetuses with labor distress and 260 without). All cases were used to test the model	ANN	Patient’s history	Fetal distress	Accuracy 86%, positive predictive value 53%, negative predictive value 92%	NA
Gao et al. (2019) [79]	United States	45,858	Electronic health record data of pregnant women were collected retrospectively. All cases were used to test the model. (NA if retrospectively or prospectively collected).	Observational medical outcomes partnership	Patient’s history, internet	Severe maternal morbidity.	M0 AUC 0.790, M1 AUC 0.919, M3 AUC 0.937	NA

Despite the group’s heterogeneity, the previous groups’ predictors were also used. For example, the studies described the prediction of pregnancy outcomes among women affected by SLE using a binary logistic regression model [75], and they described the prediction of congenital heart disease (CHD) among pregnant women [44], the frequency of 27 potential risk factors related to pregnancy and the perinatal period [77], and the identification of patients with the risk of fetal distress in labor using ANNs [78]. The application of ML was used to identify severe maternal morbidity (SMM) and relevant risk factors from electronic health records (EHRs) [79].

There were only a few of the most relevant prospective AI models tested on the representative samples [35,36,37,42,44,50,71]. Therefore, Neocleous et al. conducted three prospective studies on the prediction of fetal aneuploidy [35,36,37]. In those studies, the sample sizes were 51,208; 72,654; and 123,329, respectively. The most widely used ANN model (fully connected multilayer feed-forward structure—FCMLFF) was subsequently trained and tested to predict trisomy 21 and other aneuploidies. The model’s accuracy was assessed to be between 94 and 100% for trisomy 21 and 80–96% for other aneuploidies [35,36,37].

For preeclampsia assessment, Jhee et al. conducted a prospective study of 11,006 pregnant patients [42]. The machine learning model—decision trees (DTs), the naïve Bayes classification (NBC), the support vector machine (SVM), the random forest (RF), and stochastic gradient boosting (SGB) were subsequently trained and used. The predictive value of the DT was 84.7%, and it was 89.9% for the NBC, 89.2% for the SVM, 92.3% for the RF, and 97.3% for SGB [42].

Li et al. conducted a prospective study to predict the fetal growth restriction on a sample of 215568 pregnant women. The ML technique—the support vector machine (SVM)—and the random forest (RF) were trained and tested in comparison to logistic regression (LR) and sparse LR. The accuracy of the SVM was 92.4%, and it was 43.7% for C4.5, 61.2% for RF; 94.5% for sparse LR; the AUC for LR was 0.6 with 93% accuracy of the model [44].

Koivu et al. conducted a prospective study by including 16,340,661 pregnant women to predict early and late stillbirth as well as preterm delivery. The ANN models, deep NN (neuronal network), SELU N (scaled exponential linear units network), and LGBM (the light GBM—gradient boosting decision tree), were subsequently trained and tested in comparison to LR. The predictive values of those models measured with an AUC in early stillbirth prediction were 0.73 for deep NN, 0.75 for SELU N, and 0.75 for LGBM compared to 0.73 for LR. In late stillbirth prediction, the AUC values were 0.57 for deep NN, 0.59 for SELU N, and 0.6 for LGBM compared to 0.58 for LR. The values of preterm delivery prediction measured with AUC were 0.66 for deep NN, 0.67 for SELU N, and 0.67 for LGBM compared to 0.64 for LR [50]. In their study, Catley et al. showed the predictive accuracy of PTB occurrence (AUC 0.73) in a prospective cohort (N = 19710) using the Neural Network Toolkit [71].

## 4. Discussion

All of the presented works showed the potential of AI methods for improving risk assessment and predicting adverse perinatal outcomes. We found that AI techniques had high prediction values established at around 80–90%, which were better in comparison to logistic regression methods. However, this systematic review did not distinguish the best AI method, and further prospective studies should be performed. We suppose that there are two reasons for this. First, every perinatal complication has different risk factors and occurrences [80,81,82,83], and a comparison of each led to biases of the heterogeneity of the results. The other reason is that most of models were tested on small groups or were not proven prospectively.

Risk factors for PTO, primarily sociological and clinical indicators, the mother’s health, problems during pregnancy, comorbidities, laboratory values, and fetal monitoring parameters, were considered. However, many more characteristics might be taken into account to measure the overall risk of a pregnancy. Doing so would increase the model’s representativeness and accuracy, but it can be difficult to judge how beneficial this would be. Immunohistochemical and genetic predictors are two of the components that are typically not taken into account. Medical specialties like immunology and genetics have advanced extremely quickly, and these fields of study have produced the orphan molecules responsible for many illnesses. Therefore, combining genetic and immunohistochemistry predictors with the previously discussed socioeconomic, laboratory, and medical history components in AI models may improve their ability to predict outcomes.

Study construction and sample size are the most significant sources of bias in the included studies. There is a majority of retrospective studies or studies where the prospective or retrospective character is not indicated. Moreover, studies where the prospective analysis of patient records was made were often conducted on small samples. As a result, only a few studies provide robust evidence of AI accuracy [35,36,37,42,44,50,71]. Based on those studies, ANN models seem appropriate for extensive patient data in APO prediction.

The usage of AI in obstetrics is not close to being a gold standard. Therefore, this summary of AI application opportunities seems crucial to show the unused potential of these methods. The authors see the opportunity for AI application in daily routine medical challenges. For example, the AI techniques’ predictive value could be used during the first assessment of a pregnant woman or even when planning the pregnancy or on the perinatology department to determine the delivery timing. Gathering the potential risk factors and using trained and validated software could lead to very early diagnosis of complications or even their prevention, for example, in patients with GDM during pregnancy [52,53]. Today’s decisions are made according to the experience and knowledge of the medical practitioners whose input in medical diagnoses and procedures is not questionable. Nevertheless, human brains are not able to proceed large prospective study data to calculate the exact risk factor in every medical case. We supposed that the digital software could help to increase medical condition prediction.

There are reports of AI applications using mobile software [84,85]. Telemedicine was applied to inform patients about their condition and gestational diabetes state. AI tools for patient examination, hospital monitoring, or everyday clinical routine could improve healthcare results. Pregnant women often underestimate their health status and do not report everything to their medical care providers [86,87]. Combining AI and telemedicine software could help medical care providers assess real-time risks and threats to pregnant women, as it works, for example, in cardiology or diabetology [88,89,90]. As previously mentioned, such information could avoid many unwanted adverse medical conditions.

This study had several limitations. One of them was a problem with synthesis because of the heterogeneity of the reported results. Therefore, several deviations into the groups were made to approximate the similar outcomes of the assessed studies. Unfortunately, this was insufficient because of methodological differences and significantly different AI methods. As a result, only AI applications with a calculated LR were quantitively synthesized. Moreover, the heterogeneity of the APO prediction reporting could have an impact on reported results. The detailed assessment of each AI method could provide the readers with the information needed to apply the AI methods in praxis. Nevertheless, there were not enough studies on the same quality level (prospective with large patient groups) according to the same complication. Therefore, more prospective studies could conclude the best AI method.

The strength of this study was its novel character. There are no published studies comparing the usage of artificial intelligence methods to predict perinatology complication occurrences.

This systematic review described AI’s application in predicting pregnancy complications, which is one of this study’s biggest strengths. It gave a broad overview of the different factors and diseases to which AI can be applied [1,16].

## 5. Conclusions

The application of AI methods as a digital software can help medical practitioners in their everyday practice in pregnancy risk assessment. Decision making supported by technology could eliminate the mistakes made because of the imperfect human brain. Based on published studies, models that used ANNs and were tested on large prospective data could be applied in APO prediction. Nevertheless, further studies could identify new methods with even better prediction potential.

## Figures and Tables

**Figure 1 healthcare-10-02164-f001:**
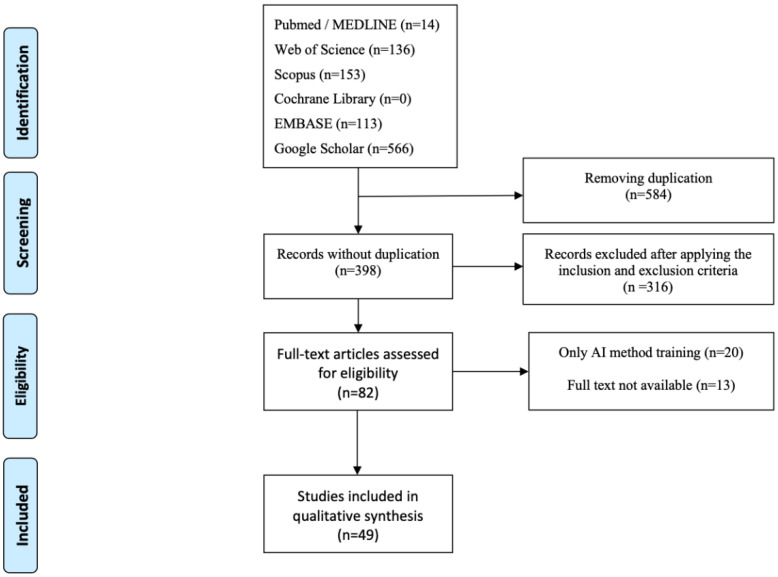
PRISMA systematic-review flow diagram.

## Data Availability

Not applicable.

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
