# Peer review of "Application of Artificial Intelligence in Screening for Adverse Perinatal Outcomes—A Systematic Review"

_healthcare, 2022, doi:10.3390/healthcare10112164_

Round 1
Reviewer 1 Report
1. Even though the authors have systematically filtered and extracted relevant literatures in Materials and Methods section, they have not discussed the related methods and applications in detail in each literature category in Results section nor Discussion section, thus, as a review paper, its contribution may be insufficient.
2. In Discussion section, the research goals and contributions of the current paper cannot be seen from their statements.
3. This paper does not show the substantial contributions, moreover, its grammar and composition are somewhat illogical.
Author Response
- Even though the authors have systematically filtered and extracted relevant literatures in Materials and Methods section, they have not discussed the related methods and applications in detail in each literature category in Results section nor Discussion section, thus, as a review paper, its contribution may be insufficient.
- Thank you for your remark. You are correct that an assessment of the methods would be very helpful for performing future studies. Nevertheless, the aim of the study was not to assess the methods themselves but their applications and efficiency on pregnancy outcomes.
- In Discussion section, the research goals and contributions of the current paper cannot be seen from their statements.
- Thank you very much for your remark. We tried to rewrite the discussion section and underline the significance of our work, discussing the results.
- This paper does not show the substantial contributions, moreover, its grammar and composition are somewhat illogical.
- Thank you for this comment. We make several changes to improve the quality of the manuscript and performed the language corrections from a native speaker. We hope that this manuscript version will be more matching for publication in healthcare.
Reviewer 2 Report
Dear authors,
Your paper could be considerably better in improving the presentation. At present, your paper is just a compilation of references and tables. Your research field is too broad, and you cannot focus on specific and interesting results from the literature. You have then to narrow your meta-analysis and perhaps describe better the papers you refer to.
On the other hand, it would be necessary that you proofread your manuscript by a professional. You will find attached your document with all my question marks.
With my kind regards

Author Response
Dear authors,
Your paper could be considerably better in improving the presentation. At present, your paper is just a compilation of references and tables. Your research field is too broad, and you cannot focus on specific and interesting results from the literature. You have then to narrow your meta-analysis and perhaps describe better the papers you refer to.
- Thank you very much for this comment. You are undoubtedly right about the broad of the study. Nevertheless, it was our aim to compare all methods used in APO prediction. The specific assessment of each complication and each method would be challenging to perform as a lot of information would be missed. For example, some methods will be discussed at al.
On the other hand, it would be necessary that you proofread your manuscript by a professional.
- Thank you very much. The AI professionals, as well as professional gynecologists, were involved in the manufacturing of the manuscript, and narrative speaker corrections were made.
You will find attached your document with all my question marks.
- Thank you very much. According to your remarks, the manuscript was corrected.
Reviewer 3 Report
The authors present the application of artificial intelligence in screening for adverse perinatal outcomes. In general, the main conclusions presented in the paper are supported by the figures and supporting text. However, to meet the journal quality standards, the following comments need to be addressed.
· Abstract: Should be improved and extended. The authors talk lot about the problem formulation, but novelty of the proposed model is missing. Also provided the general applicability of their model. Please be specific what are the main quantitative results to attract general audiences.
· The introduction can be improved. The authors should focus on extending the novelty of the current study. Emphasize should be given in improvement of the model (in quantitative sense) compared to existing state-of-the art models.
· More details about network architecture and complexity of the model should be provided.
· what about comparison of the result with current state-of-the art models? Did authors perform ablation study to compare with different models?
· What are the baseline models and benchmark results? The authors may compared the result with existing models evaluated with datasets
· Conclusion parts needs to be strengthened.
· Please provide a fair weakness and limitation of the model, and how it can be improved.
· Typographical errors: There are several minor grammatical errors and incorrect sentence structures. Please run this through a spell checker.
Author Response
The authors present the application of artificial intelligence in screening for adverse perinatal outcomes. In general, the main conclusions presented in the paper are supported by the figures and supporting text. However, to meet the journal quality standards, the following comments need to be addressed.
- Thank you for your comment. We hope that the current version of the manuscript will suite the journal standards.
- Abstract: Should be improved and extended. The authors talk lot about the problem formulation, but novelty of the proposed model is missing. Also provided the general applicability of their model. Please be specific what are the main quantitative results to attract general audiences.
- Thank you for your remark. The abstract and was revised according to your suggestion. We wanted to compare application of AI models in perinatology mot the methodology itself but the outcomes.
- The introduction can be improved. The authors should focus on extending the novelty of the current study. Emphasize should be given in improvement of the model (in quantitative sense) compared to existing state-of-the art models.
- Thank you for your remark. The introduction was revised according to your suggestion.
- More details about network architecture and complexity of the model should be provided.
- Thank you for your remark. Nevertheless, the aim of the study was not to provide the readers with information about the construction of models itself but about their application. Additional information about the construction of the models would probably make the manuscript less transparent.
- what about comparison of the result with current state-of-the art models? Did authors perform ablation study to compare with different models?
- Thank you for your remark. We did not use any of the models in our study. We are supposed to do it in the next step. First, we compare in this study the AI methods used on pregnant women. Then we are supposed to use it on prospective data.
- What are the baseline models and benchmark results? The authors may compared the result with existing models evaluated with datasets
- Thank you for this comment. Nevertheless, it was not the aim of the study to compare the construction of the model but their application.
- Conclusion parts needs to be strengthened.
- Thank you. We changed the conclusion section to mention the most significant information of the manuscript.
- Please provide a fair weakness and limitation of the model, and how it can be improved.
- No specific model was described in the study. We were supposed to assess the effectivity of each and compare them according to pregnant patients and APO occurrence.
Authors had
- Typographical errors: There are several minor grammatical errors and incorrect sentence structures. Please run this through a spell checker.
- Thank you very much. Native speaker corrections were made.
Reviewer 4 Report
My comments are as follows.
1. Please mention the advantages of your review.
2. why do authors collect the data and prepare a review paper, Is the published work something missing? Justify in detail.
3. I think it will be better to extend the introduction section, and only cite the related papers.
4. for some tables I think authors should write some theoretical detail for the reader's group.
5. authors should revise the abstract, and write it in a good way.
6. the conclusion section is not good. please re-write it with proper detail.
7. In the discussion section, it will be better to show the significance.
Author Response
- Please mention the advantages of your review.
- The strength of the study is the novel character. There are no published studies comparing usage of artificial intelligence methods to predict perinatology complications occurrence. Application of AI methods as a digital software could help medical practitioners in their everyday practice in pregnancy risk assessment.
- why do authors collect the data and prepare a review paper, Is the published work something missing? Justify in detail.
- The study compared the present results for applying artificial intelligence methods in obstetric pregnancy risk assessment and predicting adverse pregnancy outcomes.
- 3. I think it will be better to extend the introduction section, and only cite the related papers.
- Thank you very much. The relevant information was added to the introduction section, and the references were changed.
- for some tables I think authors should write some theoretical detail for the reader's group.
- Thank you for your remark. We tried to explain the tables in the manuscript text in connected with the table.
- authors should revise the abstract, and write it in a good way.
- Thank you very much. The abstract was rewritten.
- the conclusion section is not good. please re-write it with proper detail.
- Thank you. We changed the conclusion section to mention the most significant information of the manuscript.
- In the discussion section, it will be better to show the significance.
- Thank you very much for your remark. We tried to underline the significance of our work in the discussion section.
Round 2
Reviewer 1 Report
The revised version is too chaotic to read. Apart from the revision of grammar and vocabulary, other sections such as methodology does not seem much improvement.
Author Response
Thank you for your comment. Additional changes were made.
Reviewer 2 Report
Dear Authors,
You have substantially improved your manuscript regarding the English wording and sentencing. I have spotted some typos. Just have a look at the attached file.
Nevertheless, I remain a little bit deceived by the overall article. From my point of view, the subject is too broad, and you cannot zero in on your aims. Your conclusions are apparent and say nothing new.
Yours faithfully

Author Response
Thank you for your remark. You are undoubtedly correct. Nevertheless, to perform more specific systematic reviews more studies are required.
CHanges you suggested were made.
Reviewer 3 Report
Revised manuscript is suitable for publication
Author Response
Thank you for your review.
Reviewer 4 Report
Accepted
Author Response
Thank you for your review